# Potential geographic distribution of relict plant *Pteroceltis tatarinowii* in China under climate change scenarios

**Jingtian Yang[1], Pan Jiang[2], Yi Huang[1], Yulin Yang[3], Rulin Wang[1], Yuxia Yang[4]\***

**1** Ecological Security and Protection Key Laboratory of Sichuan Province, Mianyang Normal University, Mianyang, PR China, **2** College of Environment and Resources, Southwest University of Science and Technology, Mianyang, PR China, **3** Sichuan Academy of Forestry Sciences, Chengdu, PR China, **4** Sichuan Provincial Key Laboratory of Quality and Innovation Research of Chinese Materia Medica, Sichuan Academy of Traditional Chinese Medicine Sciences, Chengdu, PR China

\* yangyuxia-7@163.com

**Data Availability Statement:** All occurrence data are available from Figshare at: https://dx.doi.org/10.6084/m9.figshare.17257208.

**Funding:** JY and YH have been funded by the Scientific research initiation project of Mianyang

## Abstract

*Pteroceltis tatarinowii* (Pteroceltis: Ulmaceae) is a deciduous tree that has a cultivation history of more than 2000 years in China. As an excellent afforestation tree species and rare and endangered tertiary relic plant, *P. tatarinowii* has high ecological protection value. Due to the forest destruction caused by predatory logging and natural environmental factors, the population of *P. tatarinowii* in China has decreased significantly. In this study, the potential geographical distribution of *P. tatarinowii* in China under climate change was predicted using MaxEnt model and ArcGIS based on 223 effective distribution points of *P. tatarinowii* and 11 environmental variables. The results showed that: (1) the prediction accuracy of MaxEnt model was extremely high, and the areas under curve (AUC) value of the training data was 0.936; The area of the potential suitable habitat area of *P. tatarinowii* under current climate condition was $180.84 \times 10^4$ km$^2$, and mainly located in the central and southeast regions of China. (2) The domain environmental variables affecting the potential geographical distribution of *P. tatarinowii* were min temperature of coldest month (12.1~22.7°C), isothermality (26.6~35.8), mean diurnal range 6.9~9.3°C and precipitation of wettest month (189.5~955.5 mm). (3) In 2050s and 2070s, compared with current ($4.19 \times 10^4$ km$^2$), the area of highly suitable habitat will increase by 0.2%-0.3% (RCP2.6) and 1.22%-3.84% (RCP8.5) respectively. while the poorly, moderately and total suitable habitats will decrease. The gravity center of *P. tatarinowii* showed a trend of migration to higher latitudes and northern regions in the future. These results will provide theoretical basis for cultivation management and resource protection of *P. tatarinowii*.

## 1 Introduction

Among the top ten global environmental problems, climate change has been listed as the primary problem and has attracted more and more attention [1–3]. Responses and feedbacks generated by terrestrial ecosystems have become one of the key research space and focal issues

Normal University (QD2019A13) and the Open Project from the Ecological Security and Protection Key Laboratory of Sichuan Province (ESP1608 and ESP1801).

**Competing interests:** The authors have declared that no competing interests exist.

under global climate change [4–6]. Back in the early 1900s, Grinnell pointed out that climate plays an important role in determining the distribution of species. Plants, which serves as an important indicator, will not only be affected by climate change, but also affect the climate through negative feedback [6, 7]. A research team has already found that global warming could cause plants to migrate to higher altitudes and latitudes [8]. Climate change will change the distribution of species and intensifies habitat fragmentation [9]. The reduction of population and genetic diversity will lead to plant migration, endangered or even extinction, especially those species with narrow niche [10, 11]. The fifth assessment report of the Intergovernmental Panel on Climate Change (IPCC, 2014) pointed out that the current global average temperature has increased by 1˚C versus that before the industrial revolution, the global average temperature in 2016–2035 may increase by 0.3–0.7˚Cversus that in 1986–2005 [12]. Studies have shown that concentration increase of greenhouse gas emissions may result in a sustained increase [13]. Changes in the spatial and temporal pattern of climate may lead to changes in the geographical distribution of rare and endangered species, thus threatening their original habitats [14, 15]. Therefore, a systematic verification of the geographical distribution pattern of rare and endangered species under climate change scenarios can effectively protect the habitat of rare and endangered species as well as the authenticity of ecological system [16].

Species distribution models (SDMs) have been widely adopted to study the impacts of individual ecology and climate change on the potential geographic distribution of species [17, 18]. Such models are also increasingly playing an important role in inferring the potential geographic distribution of species by using changes of environmental variables, and they have been widely used to simulate the potential geographic distribution of rare and endangered species in recent years [19]. At present, commonly used SDMs include BIOCLIM, CLIMEX, DOMAIN, GARP and MaxEnt among others [20, 21]. MaxEnt can find out the maximum entropy of species distribution probability through known species distribution data and environmental factors, emphasize the impact of human interference factors, abiotic environmental factors and biological factors on species distribution in the simulation process, and has the advantages of requiring only 'no-absence points', stable prediction results, regular program, reducing the probability of over fitting, etc [22, 23]. MaxEnt's prediction results are more delicate and more suitable for the prediction of endangered protected plants with limited distribution data and narrow niche [24, 25], such as *Zelkova schneideriana* [26], *Prunus africana* [27], *Helianthemum songaricum* [28] and *Alsophila spinulos* [29].

*Pteroceltis tatarinowii* Maxim. (Pteroceltis: Ulmaceae) is a deciduous tree that has a cultivation history of more than 2000 years in China [30, 31]. As an excellent afforestation tree species and rare and endangered tertiary relic plant, *P. tatarinowii* has high ecological protection value [31, 32]. Besides, it is a unique monotypic plant with important scientific research value to study the systematic evolution of Ulmaceae. In view of the high ecological, economic and social benefits of *P. tatarinowii*, the protection of its germplasm and genetic resources has important theoretical and practical application value [32]. In recent years, due to the forest destruction caused by predatory logging and natural environmental factors, the populations of *P. tatarinowii* in China has decreased significantly, scattered in Northern Jiangsu, Shandong, North China and the southern end of Liaodong Peninsula. At present, studies on *P. tatarinowii* mainly focus on morphological development [33], systematic classification [34] and genetic diversity [35]. Research on species distribution pattern and potential suitable habitat of *P. tatarinowii* still remains vacant.

Based on the distribution data of *P. tatarinowii* and climatic variables, the spatial analysis technology of MaxEnt model and ArcGIS software were used to simulate the potential geographical distribution of *P. tatarinowii* under current climatic condition, the relationship between *P. tatarinowii* and variables was analyzed, the potential geographical distribution

under future climate change scenarios were predicted, the changes and heterogeneity of *P. tatarinowii* in different regions of China in the future were clarified (Fig 1).

## 2 Materials and methods

### 2.1 Study area

China is located in the east of Asia and the West Bank of the Pacific Ocean, with vast territory and complex and diverse terrain. Complex and diverse climate, significant monsoon climate and strong continental climate are typical characteristics of China's climate. From the perspective of climate types, the eastern part of China belongs to subtropical monsoon climate, temperate monsoon climate and tropical monsoon climate, the northwest belongs to temperate

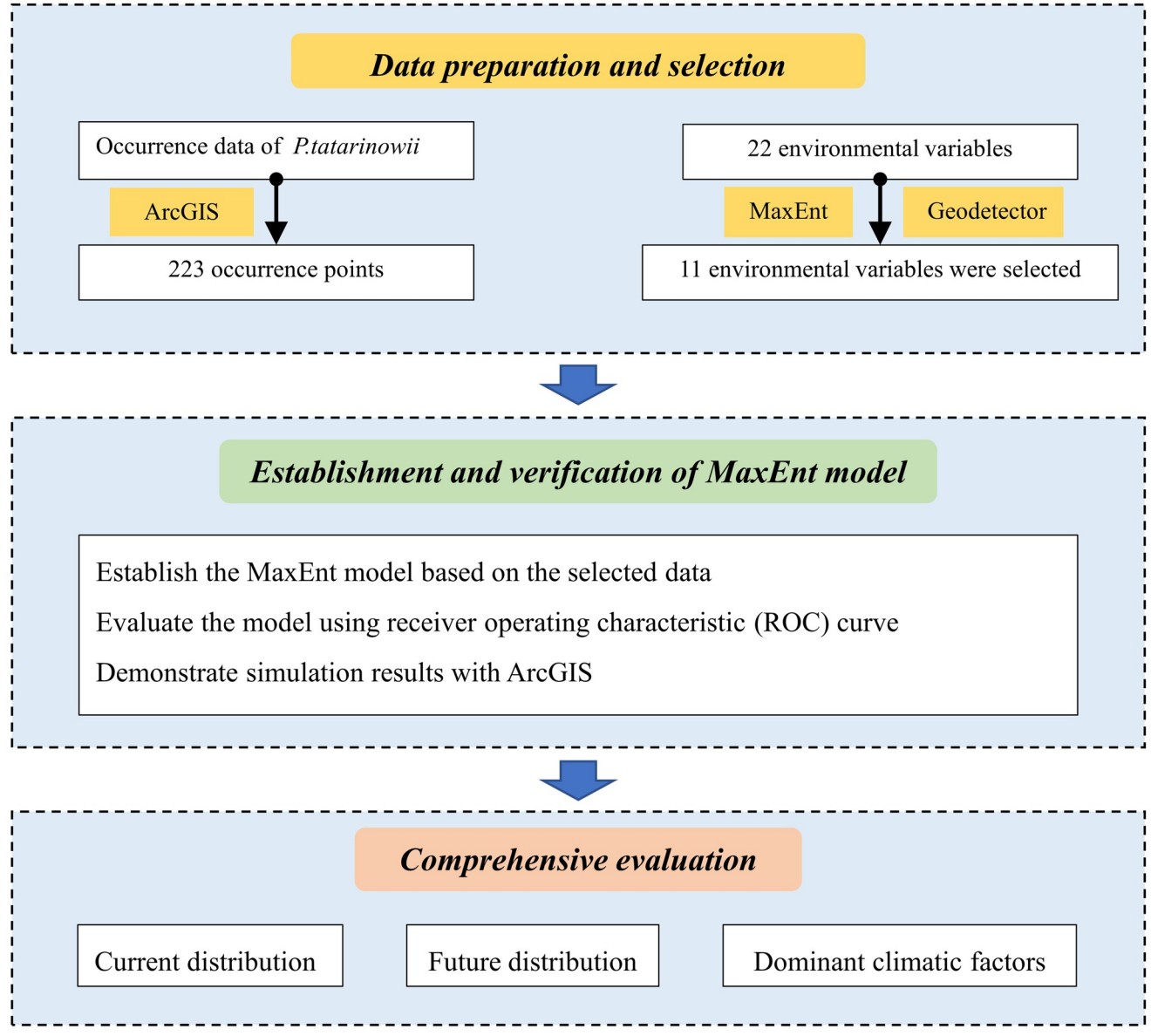

**Fig 1. Flowchart displaying the steps of the present study.**

continental climate, and the Qinghai Tibet Plateau belongs to alpine climate [36]. Second, as China has the most significant monsoon climate in the world, most parts of China have the same period of rain and heat, that is, cold and dry winter, warm, hot and rainy summer. With this complex and diverse climate, most crops, animals and plants in the world have suitable places to grow in China [37]. However, the complexity of climate leads to frequent occurrence of disastrous weather in China, such as drought, flood, cold wave and typhoon, which will have an adverse impact on the reproduction, development and distribution of animals and plants [38, 39].

## 2.2 Source of distribution records of *P. tatarinowii*

In this research, the occurrences of *P. tatarinowii* were obtained by visiting the National Plant Specific Resource Center (NPSRC, http://www.cvh.ac.cn/), the National Plant Specific Resource Center (NPSRC, Http://www.cvh.ac.cn/). The above data come from different herbarium or institutions, so there may be partial overlap in location. On the other hand, due to the different collection personnel and collection time, the accessibility of species distribution areas and human research bias, the species distribution data will be too dense in some areas. In order to eliminate the influence of this part of data on the prediction results to a certain extent, the data were screened according to the previous research methods [40–42]. Firstly, the points with overlapping positions, fuzzy location description and no clear latitude and longitude marks were removed. Second, in order to avoid over fitting, a buffer zone with each distribution point as the center and a radius of 5 km was established by using ArcGIS, and only one distribution point was reserved within 5 km [25, 27]. Finally, 223 distribution points of *P. tatarinowii* were collected (Fig 2, S1 Data).

## 2.3 Environmental variables and related data

19 bioclimtic variables with a coordinate system of WGS84 were obtained by accessing the WorldClim Database (http://www.worldclim.org//). WorldClim is a database of high spatial resolution global weather and climate data. These data can be used for mapping and spatial modeling [22, 24, 28, 29]. The database collects detailed meteorological information recorded by meteorological stations around the world from 1970 to 2000. The altitude, slope and aspect data were downloaded from the geospatial Data Cloud (https://www.gscloud.cn/).

BCC-CSM2-MR (Beijing Climate Center Climate System Model) of the Coupled Model Intercomparison Project Phase 6 (CMIP6) was selected as the future climate model. BCC-CSM2-MR, developed by China National Climate Center, has been proved to be more suitable for China's climate change characteristics and has been used in similar studies [43–45]. The Shared Socio-economic Pathways (SSPs) scenarios which can describe the prediction results of future climate change more scientifically were chosen [43, 46, 47]. Since there is a certain correlation between environmental variables, correlation analysis is required before modelling [48]. First, all variables were imported into the MaxEnt and repeated 3 times, and the variables with the contribution rate of 0 were deleted (S1 Table). Secondly, all the remained variables were selected for Spearman correlation analysis (S2 Table). When the correlation coefficient of the two variables was greater than or equal to 0.8, the one with higher contribution rate was retained. Finally 11 environmental variables were retained to build the final model (Table 1).

The 1: 16 million Chinese administrative division map was taken from the Ministry of Natural Resources of the People's Republic of China (http://bzdt.ch.mnr.gov.cn/index.html), Map

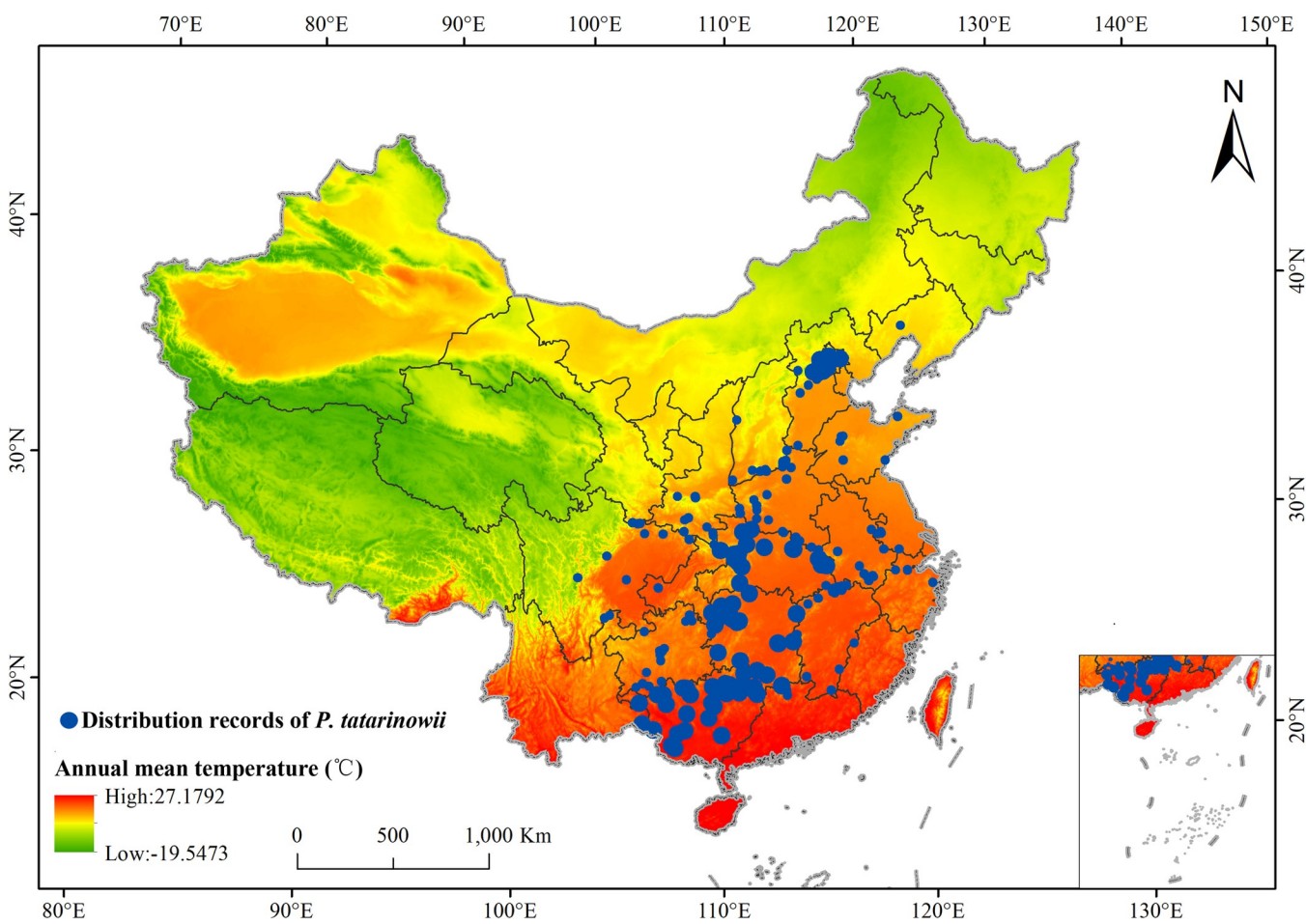

**Fig 2. The distribution records of *P. tatarinowi i*.** Larger circles indicating higher numbers of this plant, smaller circles indicating less numbers. The boundary was obtained from the Ministry of Natural Resources of the People's Republic of China (http://bzdt.ch.mnr.gov.cn/index.html), Map review number: GS(2016)2923.

review number: GS(2016)2923. MaxEnt (V3.4.1) was downloaded from the website of American Museum of Natural History (http://biodiversityinformatics.amnh.org/open source/maxent/). The ArcGIS software version used in this research was version 10.2.

**Table 1. Environmental variables used in MaxEnt model.**

| Index | Descriptions |
|---|---|
| bio2 | Mean diurnal range (Mean of monthly (max temp—min temp)) |
| bio3 | Isothermality (Bio2/Bio7×100) |
| bio4 | Temperature seasonality (standard deviation×100) |
| bio6 | Min temperature of coldest month |
| bio10 | Mean temperature of warmest quarter |
| bio13 | Precipitation of wettest month |
| bio15 | Precipitation seasonality (Coefficient of Variation) |
| bio19 | Precipitation of the coldest quarter |
| Altitude | Altitude |
| Aspect | Aspect |
| Slope | Slope |

## 2.4 Parameters of MaxEnt model

The distribution points of *P. tatarinowii* and 11 environmental variables were imported into MaxEnt to establish the model. In the basic parameter setting options, 'Random test percentage' was set to 25 and 'Max number of background points' was set to 10000. In the advanced parameter setting options, 'Maximum iterations' was set to 500 and 'Convergence threshold' was set to 0.00001. Other parameters were selected as model default. For subsequent analysis, the functions of 'Creating response curves', 'Make pictures of predictions' and 'Do jackknife to measure variable importance' were selected in turn [23].

In this research, in order to evaluate the performance of parameter configuration, different parameter configurations were selected for trial operation so as to achieve the purpose of adjusting the optimal parameters of the model [49]. Firstly, based on the known distribution points of *P. tatarinowii* and its corresponding environmental variables, RM was set to 0.5~4 respectively. 6 feature combinations (FC) were used to optimize the model parameters and select the best parameter combination: L (linear feature); LQ (linear feature+quadratic feature), h (hinge feature), LQH (linear feature+quadratic feature+hinge feature+product feature) and LQHPT (linear feature+quadratic feature+hinge feature+product feature+threshold value feature). Finally, the RM in this research was set to step 1, with the feature combination to LQHPT, and the Jackknife method was selected to determine the importance of each variable.

## 2.5 Validation of model accuracy

The AUC was used to evaluate the prediction accuracy of the model. The closer the AUC value is to 1, the better the prediction effect of the model [23, 50]. The evaluation criteria of model simulation accuracy were divided into four grades: poor (AUC≤0.80), average (0.8<AUC≤0.90), good (0.90< AUC≤0.95) and excellent (0.95< AUC≤1.00) [41].

## 2.6 Classification of suitable habitat grades

In the output file, the average value of 10 repeats based on the existence probability logic value (P) of species was selected as the simulation result. The range of P value is 0 ~1. The larger the P value, the greater the possibility of species existence. ArcGIS10.2 software was used to convert the ASCII file into raster format, and suitable habitats were classified and visualized. Based on the P value, the Natural Breaks method was used to divide the suitable habitat into four grades, i.e. highly suitable habitat (0.5≤P≤1.0), moderately suitable habitat (0.3≤P< 0.5), poorly suitable habitat (0.1≤P < 0.3) and unsuitable habitat (0.0≤P < 0.2). The number of grids in each suitable area was counted, and the area and proportion were calculated.

Referring to Yue et al., the gravity center of suitable habitats of *P. tatarinowii* under climate change scenarios were counted by using the Zonal Geometry Tool in ArcGIS, the changes of gravity center position under scenarios were calculated [51].

## 3 Results

### 3.1 Environmental variables affecting the potential geographical distribution of *P. tatarinowii*

Based on 223 distribution points, the potential geographical distribution of *P. tatarinowii* in China was simulated by using MaxEnt. The AUC values of the training data and test data were 0.936 and 0.910, respectively, indicating that the model fitting effect was good (Fig 3).

At present, there is no unified method to determine the dominant environmental variables affecting species distribution, and the cumulative contribution rate of variables is used as the basis for selection by many researchers. The threshold of cumulative contribution rate is

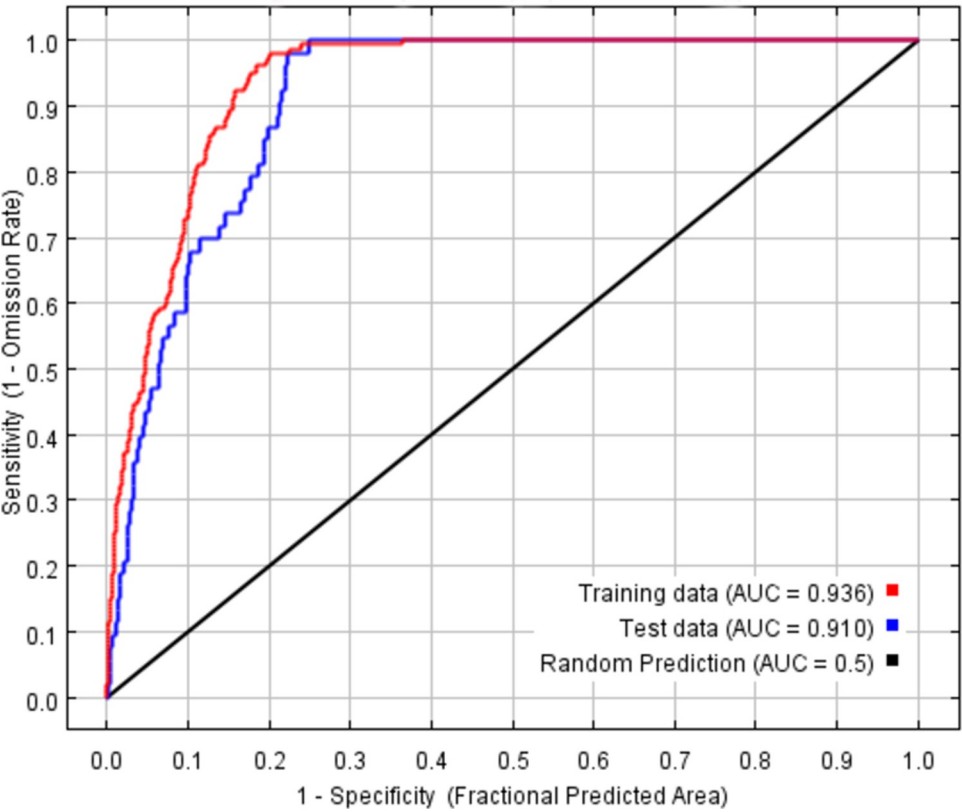

**Fig 3. Receiver operating characteristic curve (ROC) of *P. tatarinowii.***

usually selected subjectively according to the characteristics of research species and research results, so the standard is not uniform. In this research, the dominant environmental variables were selected based on contribution rate in conjunction with Jackknife method. Among the 11 environmental variables predicted by Maxent, the precipitation of the wettest month (bio13, 37.1%), the min temperature of coldest month (bio6, 28.3%), and the isothermality (bio3, 10.9%) were the variables with the highest contribution rate, with the cumulative contribution rate of 76.3% (Fig 4). The results of Jackknife test (Fig 5) showed that the precipitation of wettest month (bio13), min temperature of coldest month (bio6) and the mean diurnal range (bio2) were the variables with the greatest impact on the regularized training gain when only a single environmental variable was applied, indicating that these variables contain unique information. To sum up, the dominant environmental variables affecting the potential geographical distribution of *P. tatarinowii* were precipitation of wettest month (bio13), min temperature of coldest month (bio6), isothermality (bio3) and mean diurnal range (bio2).

The response curve of species survival probability to environmental factors in the results of MaxENT model was generally used to determine the relationship between species existence probability and environmental variables (Fig 6). Referring to the previous studies, when the existence probability of *P. tatarinowii* was greater than the threshold value of highly suitable habitat (0.5), the corresponding environmental variables interval was regarded as the suitable interval for *P. tatarinowii* to survive.

The operation results of MaxEnt model showed that when the mean diurnal range (bio2) reached 6.9˚C, the survival probability of *P. tatarinowii* reached the threshold value of highly suitable habitat (0.5). With the increase of the mean diurnal range (bio2), the existence

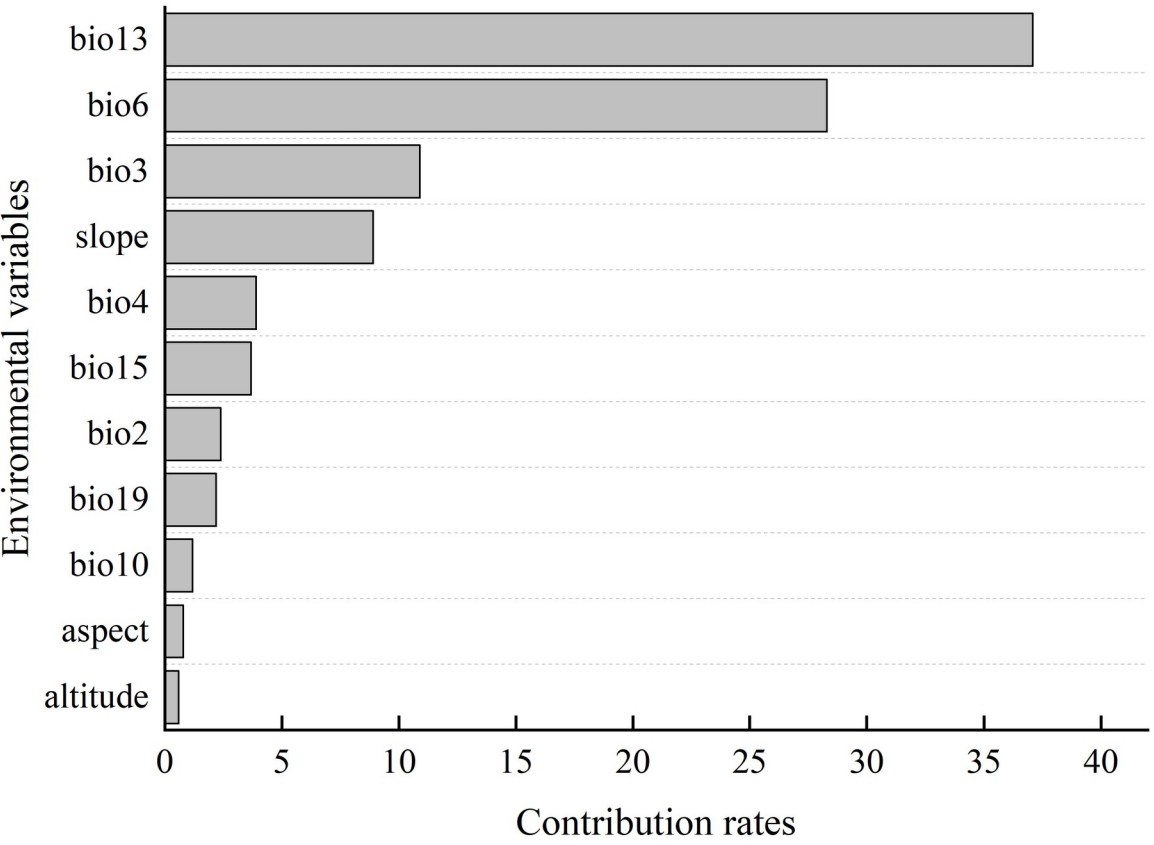

**Fig 4. Contribution rate of environmental variables on the distribution of *P. tatarinowii*.**

probability of *P. tatarinowii* reached the peak (0.62). Then the survival probability of *P. tatarinowii* decreased with the increase of the mean diurnal range (bio2). When the mean diurnal range (bio2) was 9.3°C, its survival probability was reduced to 0.5. Therefore, the suitable range of the mean diurnal range (bio2) was 6.9~9.3°C. When the isothermality (bio3) was 26.6, its survival probability reached the threshold value of highly suitable habitat (0.5). With

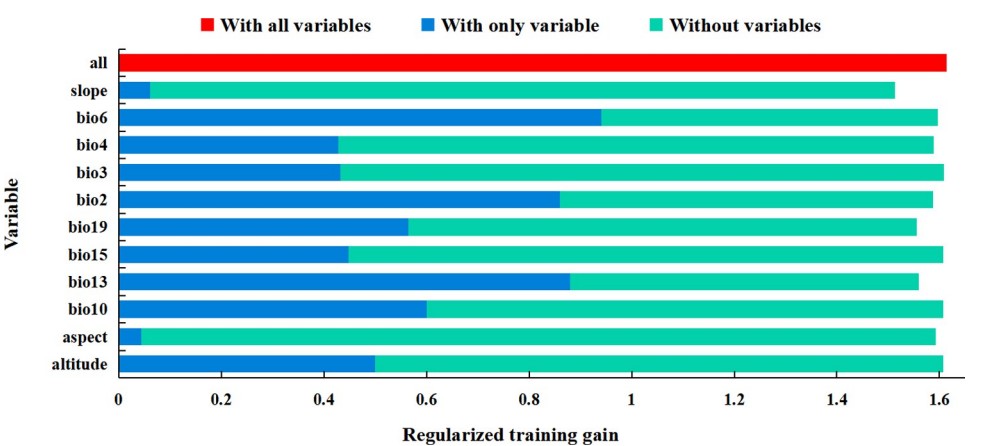

**Fig 5. The Jackknife test result of environmental variables of *P. tatarinowii*.**

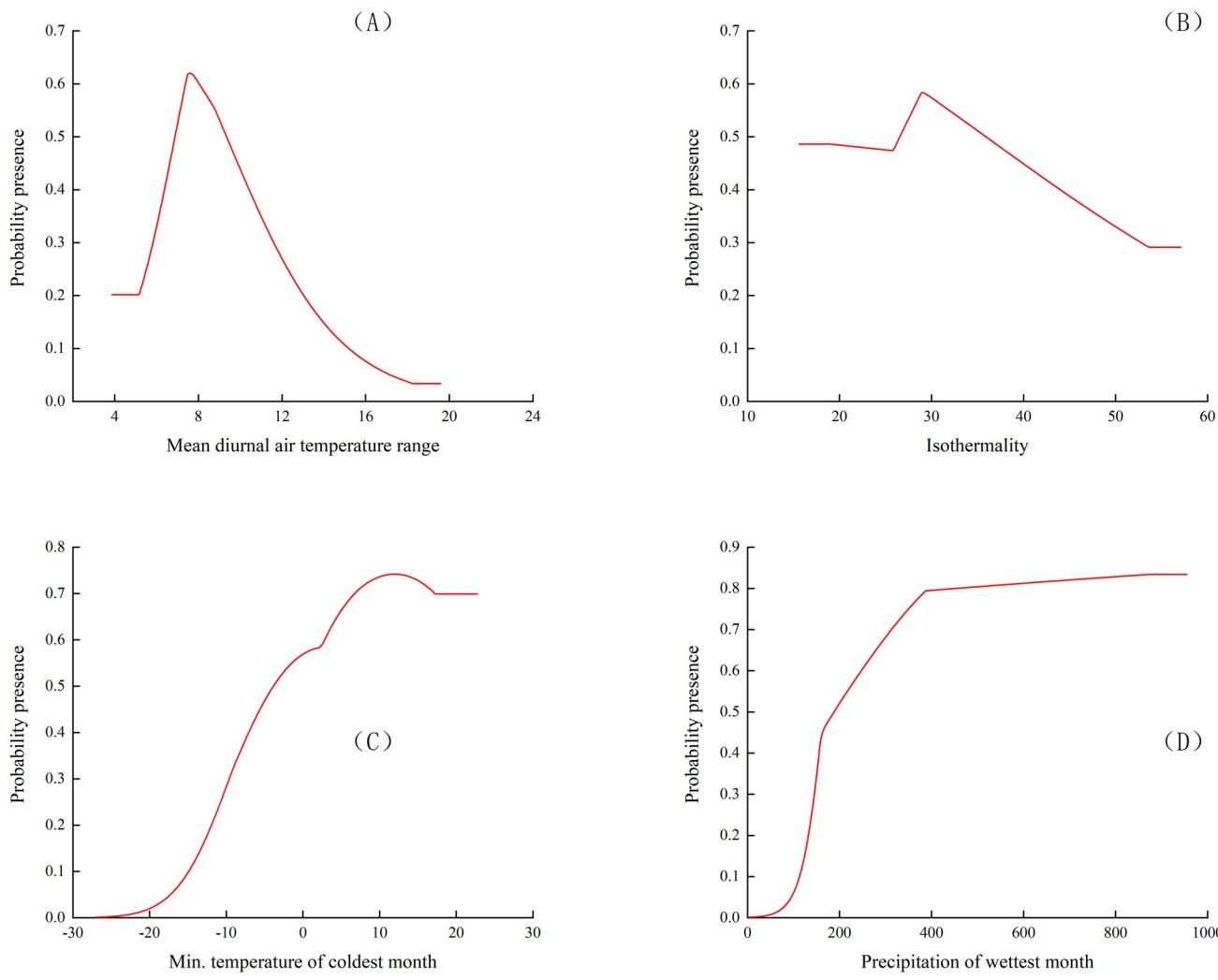

**Fig 6. Response curves of existence probability of *P. tatarinowii* for environmental variables.**

the increase of isothermality (bio3), the survival probability of *P. tatarinowii* showed an upward trend. When the isothermality (bio3) was 29.0, its survival probability reached the peak (0.58). With the continuous increase of isothermality (bio3), the survival probability of *P. tatarinowii* began to decrease. When the isothermality (bio3) was 35.8, the survival probability decreased to 0.5. Therefore, the suitable range of the isothermality (bio3) was 26.6~35.8. The survival probability of *P. tatarinowii* reached the threshold value of highly suitable habitat (0.5) when the min temperature of coldest month (bio6) reached -3.7°C. With the increase of the min temperature of coldest month (bio6), the survival probability of *P. tatarinowii* reached the peak (0.74) at 12.1°C. Then, the survival probability of *P. tatarinowii* started to decrease slightly with the increase of the min temperature of coldest month (bio6). When the min temperature of coldest month (bio6) was 22.7°C, the survival probability of *P. tatarinowii* decreased to the lowest value. Therefore, the suitable range of min temperature of coldest month (bio6) was 12.1 ~22.7°C. When the precipitation of wettest month (bio13) reached 189.5 mm, the survival probability of *P. tatarinowii* reached the threshold value of highly suitable habitat (0.5). With the increase of the precipitation of wettest month (bio13), the survival

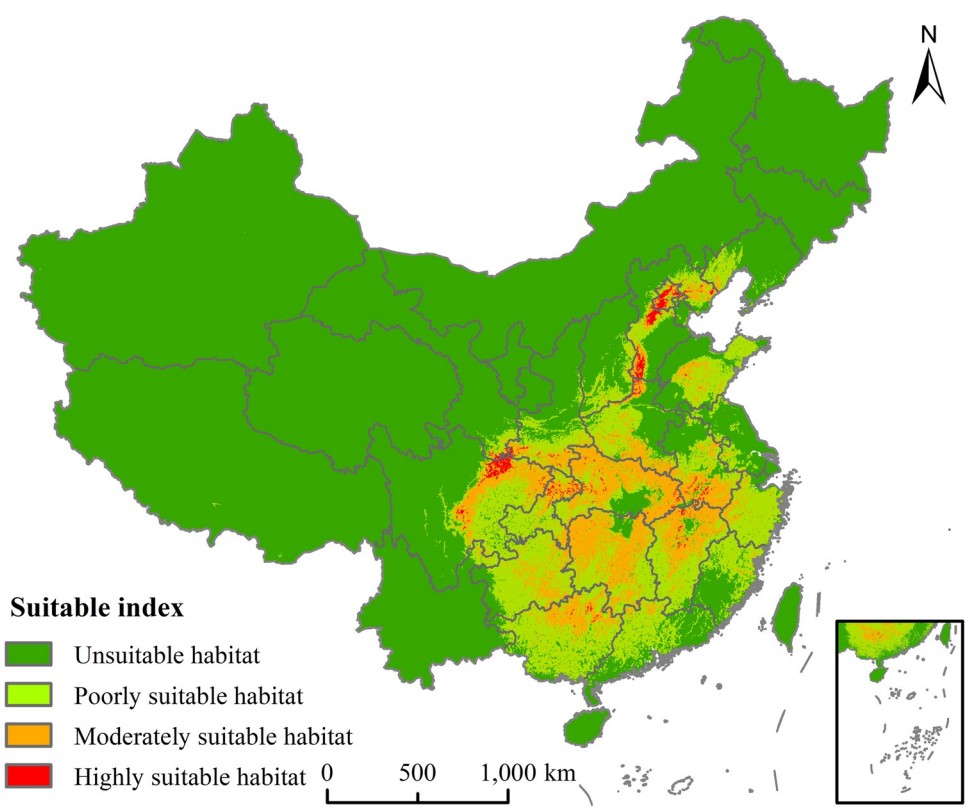

**Fig 7. Potential geographical distribution of *P. tatarinowii* under current climate conditions.** This map was made in ArcGIS 10.2 using the resulting rasters produced by MaxEnt. The boundary was obtained from the Ministry of Natural Resources of the People's Republic of China (http://bzdt.ch.mnr.gov.cn/index.html), Map review number: GS (2016)2923.

probability of *P. tatarinowii* showed an upward trend. When precipitation of wettest month (bio13) was 955.5 mm, the survival probability reached its peak (0.83). Thus, the suitable range of the survival probability was 189.5 ~955.5 mm. The suitable range of temperature and precipitation variables affecting the survival probability of *P. tatarinowii* was relatively narrow, indicating that its requirements for temperature range and precipitation of was higher.

## 3.2 Potential geographical distribution of *P. tatarinowii* under current climatic condition

The potential geographical distribution of *P. tatarinowii* under current climate condition was simulated by using MaxEnt model (Fig 7). Among the existence probability logical values of 96 effective distribution points of *P. tatarinowii*, the highest logical value was Fangshan District, Beijing (0.96), and the lowest was Yuxian, Zhangjiakou City, Hebei Province (0.01), with the average of 0.52.

The area of the highly suitable habitat of *P. tatarinowii* was $4.19 \times 10^4$ km$^2$, accounting for 0.44% of the total land area of China (Table 2), mainly located in northern Sichuan, southern Shaanxi, most of Beijing, central and southern Hebei, northern Henan and northeastern Chongqing. The area of the moderately suitable habitat was $56.36 \times 10^4$ km$^2$, accounting for 5.87% of the total land area of China (Table 2), mainly distributed in most of Hubei, central and Northern Hunan, northern Jiangxi, southern Anhui, southern Henan, northern Guangxi, central and northeastern Sichuan, central Guizhou, southern Shaanxi and western Shandong.

Table 2.  Suitable areas for *P. tatarinowii* under climate change scenarios ($10^4$km$^2$).

| Period | Highly suitable habitat | Moderately suitable habitat | Poorly suitable habitat | Total suitable habitat |
|---|---|---|---|---|
| Current | 4.19 | 56.36 | 120.29 | 180.84 |
| 2050, SSP1-2.6 | 6.19 | 8.39 | 59.69 | 74.27 |
| 2070, SSP1-2.6 | 7.06 | 14.25 | 66.59 | 87.9 |
| 2050, SSP5-8.5 | 11.67 | 26.55 | 76.39 | 114.61 |
| 2070, SSP5-8.5 | 41.04 | 40.56 | 92.79 | 174.39 |

In summary, the area of the total suitable habitat of *P. tatarinowii* was 180.84×$10^4$km$^2$, accounting for 18.8% of the total land area of China (Table 2). The potential suitable habitats of *P. tatarinowii* were distributed in the central and southwestern regions of China in a patchy manner. The highly suitable habitat was relatively narrow, and the range of its potential suitable habitat was highly consistent with the actual distribution, indicating that the simulation effect in this research was relatively accurate.

### 3.3 Prediction of the potential geographical distribution of *P. tatarinowii* under climate chang scenarios

The MaxEnt model was used to predict the potential geographical distribution of *P. tatarinowii* in China under SSP1-2.6 and SSP5-8.5 emission scenarios in 2050s and 2070s (Fig 8), The areas of suitable habitats at different levels under two different emission scenarios in 2050s and 2070s have all undergone changes in varying degrees versus those under current climate conditions (Table 2).

Under the SSP1-2.6 emission scenario in 2050s, the area of the highly suitable habitat of *P. tatarinowii* was 6.19×$10^4$ km$^2$, accounting for 0.64% of the total land area of China, 0.2% higher than that of *P. tatarinowii* under current climate condition. Under SSP1-2.6 in 2070s, the area of the highly suitable habitat of *P. tatarinowii* was 7.06×$10^4$ km$^2$, accounting for 0.74% of the total land area of China, 0.30% higher than that of *P. tatarinowii* under current climate condition. Under SSP5-8.5 in 2050s, the area of the highly suitable habitat of *P. tatarinowii* was 11.67×$10^4$ km$^2$, accounting for 1.22% of the total land area of China, 0.78% higher than that of *P. tatarinowii* under current climate condition. Under SSP5-8.5 in 2070s, the area of the highly suitable habitat of *P. tatarinowii* was 41.04×$10^4$ km$^2$, accounting for 4.28% of the total land area of China, with an increase of 3.84% compared with the current climate condition. The area of the highly suitable habitat of *P. tatarinowii* showed a trend of increase under both emission scenarios in 2050s and 2070s, with the largest increase ratio under the emission scenario of SSP5-8.5 in 2070s (Fig 8, Table 2).

Under SSP1-2.6 emission scenario in 2050s, the area of the moderately suitable habitat of *P. tatarinowii* was 8.39×$10^4$ km$^{2,}$ accounting for 0.87% of the total land area of China, 5.0% lower than that of *P. tatarinowii* under current climate condition. Under SSP1-2.6 in 2070s, the area of the moderately suitable habitat of *P. tatarinowii* was 14.25×$10^4$ km$^2$, accounting for 1.48% of the total land area of China, 4.39% lower than that of *P. tatarinowii* under current climate condition. Under SSP5-8.5 in 2050s, the area of the moderately suitable habitat of *P. tatarinowii* was 26.55×$10^4$ km$^2$, accounting for 2.77% of the total land area of China, 3.1% lower than that of *P. tatarinowii* under current climate condition, having the largest ratio of increase. Under SSP5-8.5 in 2070s, the area of the moderately suitable habitat of *P. tatarinowii* was 40.56×$10^4$ km$^2$, accounting for 4.23% of the total area of China, 1.64% lower than that of *P. tatarinowii* under current climate condition. Under future climate change scenarios, the area of the moderately suitable habitat of *P. tatarinowii* showed a decreasing trend under the two

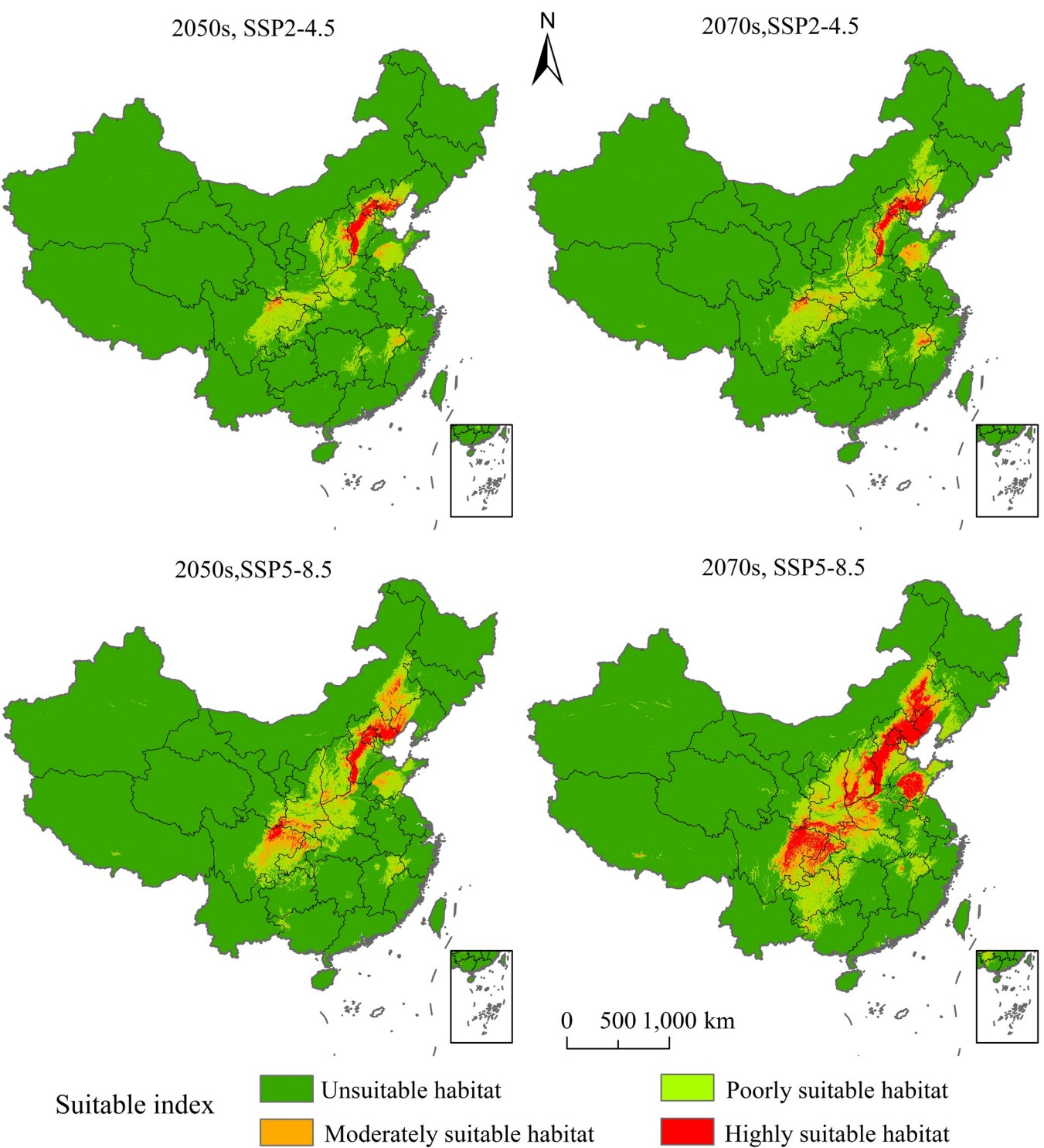

**Fig 8. Potential geographical distribution of *P. tatarinowii* under future climate change scenarios.** This map was made in ArcGIS 10.2 using the resulting rasters produced by MaxEnt. The boundary was obtained from the Ministry of Natural Resources of the People's Republic of China (http://bzdt.ch.mnr.gov.cn/index.html), Map review number: GS(2016)2923.

emission scenarios in 2050s and 2070s. It has the largest ratio of decrease under the emission scenario of SSP1-2.6 in 2050s (Fig 8, Table 2).

In summary, under future climate change scenarios, the poorly, moderately and total suitable habitats of *P. tatarinowii* showed a decreasing trend, while its highly suitable habitats showed an increasing trend, which was mainly due to the migration of poorly and moderately suitable habitats into higher and unsuitable habitats, which led to the increase of highly suitable habitats and the decrease of total suitable habitats.

Under the emission scenarios of 2050s and 2070s, the suitable habitat of *P. tatarinowii* had suffered loss in different degrees in central and southern China (Fig 9). Under SSP1-2.6 in 2050s, the loss area of suitable habitat of *P. tatarinowii* was the largest, with an area of $120.48 \times 10^4$ km$^2$. Under SSP5-8.5 in 2070, the loss area of suitable habitat of *P. tatarinowii* was the smallest, with an area of $75.98 \times 10^4$ km$^2$ (Table 3). The loss area of *P. tatarinowii* was mainly located in the southeast of China, which was mainly characterized by the loss of poorly suitable habitat and moderately suitable habitat. In some areas, the moderately suitable habitat would transformed into highly suitable habitat, leading to the increase of its potential suitable habitat and the overall migration to the north and higher latitudes. Under SSP1-2.6 in 2050s, the increasing area of suitable habitat of *P. tatarinowii* was the smallest, with an area of $14.09 \times 10^4$ km$^2$. Under SSP5-8.5 in 2070s, the increasing area of suitable habitat of *P. tatarinowii* was the largest, which was $69.58 \times 10^4$ km$^2$ (Table 3). The increasing areas of *P. tatarinowii* were mainly located in as Shanxi, Hebei and Beijing (Fig 9).

## 3.4 Moving tendency of the gravity center under future climate change scenarios

Based on the gravity center of *P. tatarinowii*'s potential suitable habitat (potential survival probability≥0.2) under current climate condition and future climate change scenarios, the moving track and change trend of its potential suitable habitat were analyzed (Fig 10). Under SSP1-2.6, the gravity center of *P. tatarinowii* would migrate from the coordinate of 30.01˚N, 112.23˚E (Current) to the coordinate of 32.73˚N, 112.19˚E (2050s), and then to the coordinate of 35.06˚N, 112.19˚E (2070s). Under SSP5-8.5, the gravity center from the coordinate of 30.01˚N, 112.23˚E (Current) to the coordinate of 35.72˚N, 112.79˚E (2050s), and then to the coordinate of 34.90˚N, 112.51˚E (2070s). Under the two emission scenarios in 2050s and 2070s, the gravity center of the potential suitable habitat of *P. tatarinowii* tended to migrate to the the north and the higher latitudes. Under SSP5-8.5 in 2050s, the migration trend of the gravity center was the most obvious (Fig 10). In conclusion, under future climate change scenarios, the potential suitable habitat of *P. tatarinowii* would migrate to the north and higher latitudes.

## 4 Discussion

### 4.1 Limitations of climatic variables on potential geographical distribution of *P. tatarinowii*

Overall accuracy, sensitivity, specificity, kappa statistics (Kappa), true skill statistic (TSS) and receiver operating characteristic (ROC) curve are the commonly used indicators to evaluate the accuracy of MaxEnt [52, 53]. ROC curve is not affected by the threshold, and is considered to be one of the best evaluation indexes at present. MaxEnt software can draw ROC curve automatically, so it is widely used to evaluate MaxEnt's performance. For example, *Li et al.* used the AUC (area under the ROC curve) to quantify the accuracy of MaxEnt model in prediction of three *Coptis* herbs in China [54]. *Liu et al.* selected ROC curve to determine the

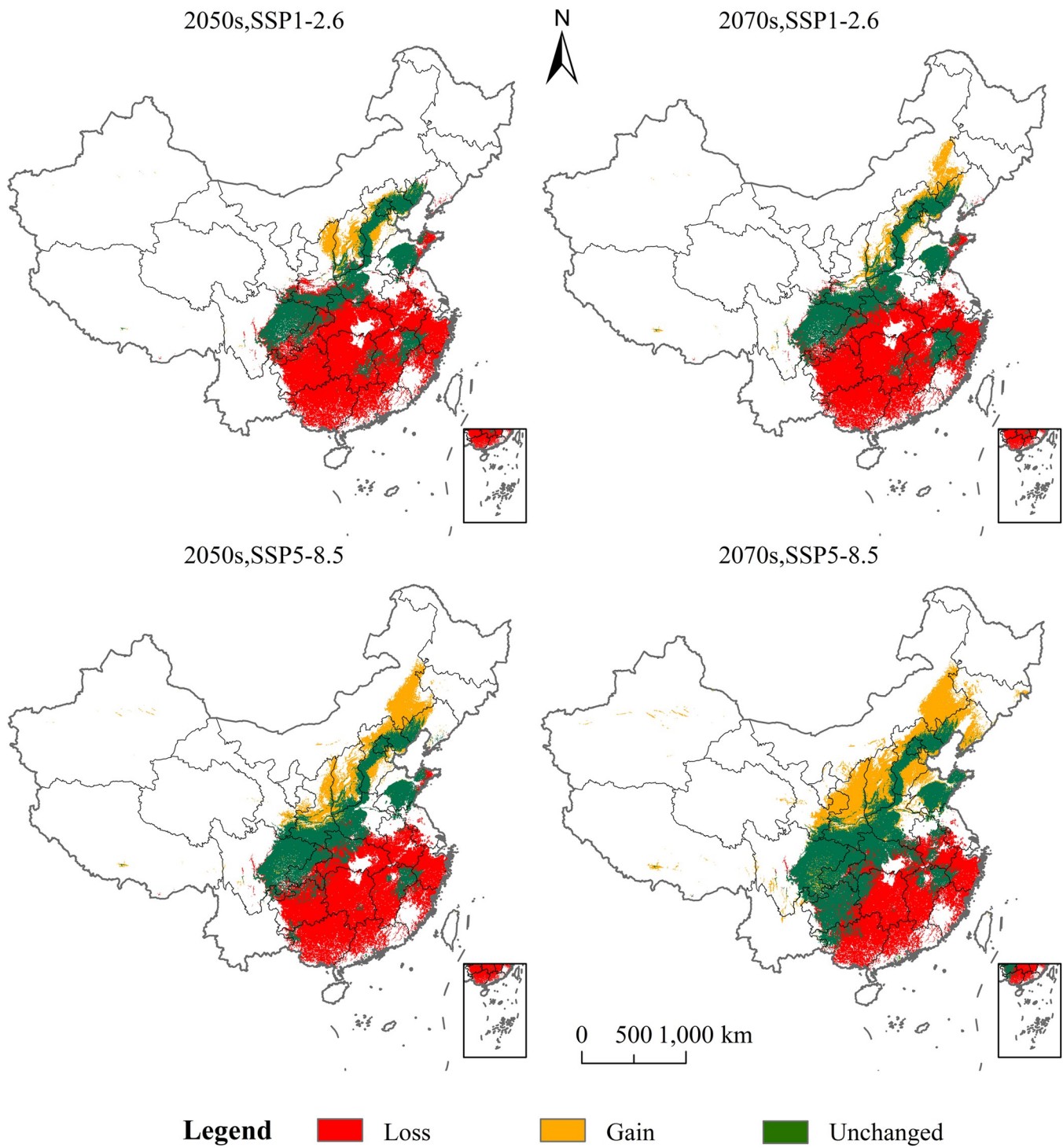

**Fig 9. Changes of potential geographical distribution of *P. tatarinowii* under future climate change scenarios.** This map was made in ArcGIS 10.2 using the resulting rasters produced by MaxEnt. The boundary was obtained from the Ministry of Natural Resources of the People's Republic of China (http://bzdt. ch.mnr.gov.cn/index.html), Map review number: GS(2016)2923.

**Table 3. Future changes of *P. tatarinowii* in suitable habitat area (10⁴km²).**

| Period | Loss | Gain | Stable |
|---|---|---|---|
| 2050, SSP1-2.6 | 120.48 | 14.09 | 60.42 |
| 2070s, SSP1-2.6 | 111.59 | 18.84 | 69.31 |
| 2050s, SSP5-8.5 | 105.80 | 39.70 | 75.20 |
| 2070s, SSP5-8.5 | 75.98 | 69.58 | 105.09 |

accuracy of MaxEnt in predicting suitable habitats for *Alnus cremastogyne* [44]. Therefore, ROC curve was used in our study.

The prediction results of MaxEnt model showed that the important environmental variables that limited the potential geographical distribution of *P. tatarinowii* were temperature variables (the min temperature of coldest month, isothermality and the mean diurnal range) and precipitation variables (the precipitation of wettest month). This research showed that with the increase of the min temperature of coldest month, the survival probability of *P. tatarinowii* increased. Related studies showed that low temperature has serious damage to *P. tatarinowii*, causing irreversible damage to its biofilm. It also showed that *P. tatarinowii* was sensitive to low temperature stress, which verified the results accuracy of this research.

Relevant research showed that *P. tatarinowii* had good adaptability to arid limestone habitat. And the distribution pattern of *P. tatarinowii* population changed with age, indicating that its distribution pattern was closely associated with its biological properties and habitat conditions [31]. *P. tatarinowii* is a sun-loving and calcium-loving plant, and its cluster distribution will continue to be maintained if it is not disturbed during the growing of seedlings or young trees [55]. When the population grows with age, its demand for light, nutrients and water increases, and the competition intensifies within or between its populations. Integrated with human interference and other factors, it will lead to self-thinning and alien thinning, causing

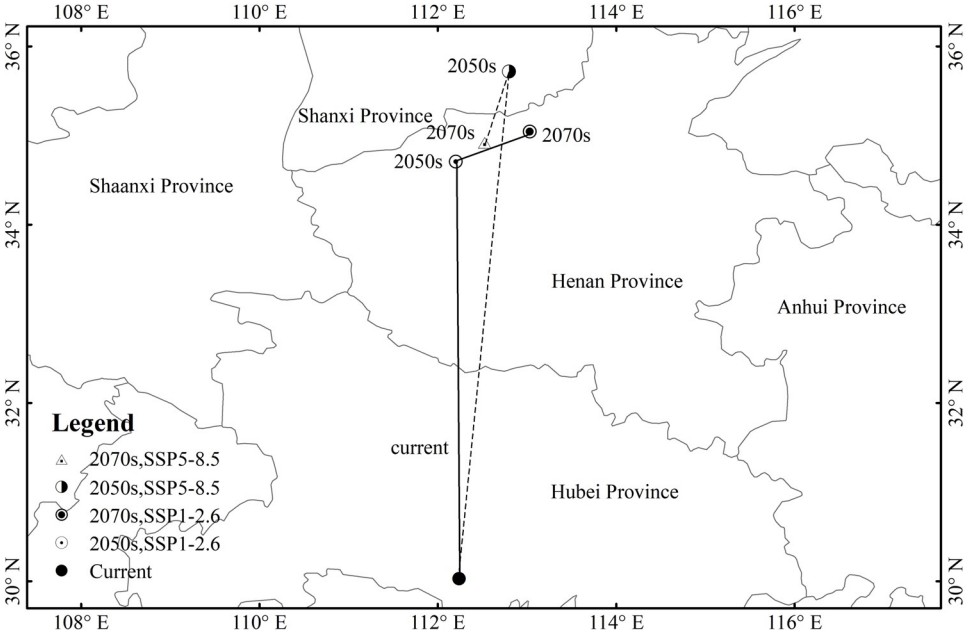

**Fig 10. Changes in gravity center of the potential suitable habitat of *P. tatarinowii* and its moving tendency under future climate change scenarios.**

reduction of the population aggregation intensity. This indicated that *P. tatarinowii* had certain requirements for water during its growth. As *P. tatarinowii* grows in relatively arid areas, precipitation plays the role of its main water source, which further verified the accuracy of the results in this research arguing that the survival probability of *P. tatarinowii* increases with the increase of precipitation. This research conducts predictions on potential geographical distribution of *P. tatarinowii* in China, and identifies the climatic variables which limits the potential geographical distribution of *P. tatarinowii*, and proposes that the expansion of the study area may change the range of environmental factors which limit the growth of *P. tatarinowii*.

Other environmental variables, such as vegetation coverage, also have certain influence on the potential geographical distribution of *P. tatarinowii*. However, this variable is not included in the simulation because it is impossible to accurately predict the change of vegetation coverage in China under climate change. As a result, some of the potential geographical distribution areas obtained in this research may be unsuitable for *P. tatarinowii* to survive, and local hydrogeological conditions must be taken into consideration in the practical application. However, the results of this study are only the first step of macro planning, and play a guiding role in the scientific management and habitat protection of *P. tatarinowii*.

## 4.2 Changes of potential geographical distribution of *P. tatarinowii* under future climate change scenarios

Shared Socio-economic Pathways (SSPs) proposed in the sixth Assessment Report (AR6) of the Intergovernmental Panel on Climate Change in 2021 shows that the range of climate warming, the increase of temperature and the increase of precipitation are becoming more prominent [56]. Thomas et al. found that 15 ~37% of the species would face the extinction risk under scenarios of moderate emission concentration in 2050s while other species would face little extinction risk, and some species even would benefit from climate warming [57]. This suggested that the impact of climate warming on the potential geographical distribution of species has two sides, and not all species are at risk of extinction or all have benefits under climate change. Our results showed that under the future climate change scenario, some regions in China would become unsuitable for the distribution of *P. tatarinowii*, while some regions would become suitable. Temperature and precipitation have an significant impact on the growth of *P. tatarinowii*, yet under SSP5-8.5 scenario in 2070s, with the increase of emission concentration, the temperature will rise, which may expand the loss scope of suitable habitat of *P. tatarinowii*. This may also be the reason for the largest areas loss of *P. tatarinowii* in this situation.

As far as the change of distribution center of gravity is concerned, Leng *et al.* (2008) used RF model to analyze the impact of climate change on the potential geographical distribution of three kinds of larches in Northeast China, showing that the potential geographical distribution of these three larches (Larix spp.) will obviously migrate to higher latitudes under future climate change scenarios [58]. Chen *et al.* studied species distribution through meta-analysis as well as other methods, and the results showed that species distribution might migrate to higher latitudes at an average speed of 16.9 km per decade [59]. Bellard *et al.* simulated the potential geographical distribution of 100 most aggressive non-native species in the world, showing that the potential geographical distribution of these species generally showed a trend of northward expansion [60]. Thuiller found that most species would migrate northward under the background of climate warming [61]. The change trend of suitable habitat of *P. tatarinowii* in the future was basically consistent with this phenomenon. The impact of climate warming on the potential geographical distribution of species is mainly manifested in the migration of the potential geographical distribution of species to higher latitude or higher altitude, as well as the

expansion and contraction of suitable areas. The potential suitable habitat of *P. tatarinowii* under future climate change scenarios would migrate to higher latitudes and northern regions, remaining consistent with this feature in this research.

## 4.3 Limitation and innovation of this research

Relict plants are of great value in the study of paleoclimate, paleogeography and floristic changes. Using SDMs to simulate the past, current and future changes of suitable habitat of relict plants is an effective method. Xu et al. simulated the change of suitable growth area of *Gymnocarpos przewalskii* from the last interglacial to current based on MaxEnt [62]. Duan et al. used MaxEnt, Bioclim and Domain to predict the potential suitable areas of *Ammopiptanthus* in the last interglacial period, the last glacial maximum, current and 2050s [63]. In this study, the distribution of *P. tatarinowii* in the past period was not simulated, which should be supplemented in the next work. Studies showed that human activities had great impact on the geographical distribution pattern of species. Sayit et al. found that after adding the variable of human activity intensity, the suitable proportion of *Calligonum mongolicum* simulated by MaxEnt decreased from 13.04% to 9.57% [64]. Cao et al. pointed out that human activities reduced the potential distribution of *Swertia przewalskii* by 32% [65]. However, the change of human activity intensity in the future is not clear, so it is not selected. Therefore, it can be inferred that the future suitable area of *P. tatarinowii* simulated in this study is wider than the actual situation.

Climate change has an indirect impact on the population and distribution characteristics of *P. tatarinowii* through directly affecting the ecosystem. Besides, the population structure and spatial distribution pattern of *P. tatarinowii* population are also affected by many factors, such as the local habitat within the population area, the biological and ecological properties of *P. tatarinowii*, as well as human interference, and so on. Under the comprehensive action of the above factors, the seedlings and young trees of *P. tatarinowii* population are few, the living environment of the existing plants is not ideal, and the population as a whole shows a declining trend. Furthermore, since most of the existing population distribution areas exist in a manner of isolated islands, resulting in population division and high inbreeding rate, causing the population further decline. Therefore, how to protect *P. tatarinowii* resources is an urgent and arduous task. Creating a protected area in the natural distribution area of *P. tatarinowii* for in-situ conservation could not only contribute to the natural regeneration of the population, but also avoid the risk of acclimatization caused by ex-situ conservation. Under the existing conditions, we may protect as many *P. tatarinowii* populations as possible, decrease human interference and reduce damage to its habitat, and appropriately increase the gene exchange among populations, so as to achieve the purpose of effective protection, and make the endangered tree species get better reproduction. This research on the potential geographical distribution pattern of *P. tatarinowii* will be helpful to understand its population dynamics and development trend. It will not only provide a theoretical basis for the adjustment when the habitat changes or the population structure changes due to human influence, but also bring important theoretical and practical significance for the exploration of endangered mechanism of its population, the rational protection and proliferation of *P. tatarinowii*. It will also provide a theoretical basis for the further study of *P. tatarinowii* population ecology.

## 5 Conclusion

The domain environmental variables affecting the potential geographical distribution of *P. tatarinowii* under current climate condition were temperature factor (min temperature of coldest month, isothermality and mean diurnal range) and precipitation variables

(precipitation of wettest month). Seasonal changes of temperature and precipitation represent changes of water and heat distribution pattern. The change of potential geographical distribution of *P. tatarinowii* was the result of multiple factors. The potential geographical distribution of *P. tatarinowii* under current climate condition was mainly in the central and southeast regions of China. Under the low and high emission scenarios in 2050s and 2070s, the highly suitable habitats showed a trend of increase. The poorly, highly and total suitable habitats showed a trend of decrease. Potential suitable habitats tended to migrate to higher latitudes and northern regions. By studying the potential geographical distribution pattern and dynamics law of *P. tatarinowii*, this paper summarized the change trend of *P. tatarinowii's* distribution pattern, and put forward scientific protection strategies.

## Supporting information

**S1 Data. Occurrence records of *Pteroceltis tatarinowii* with latitude and longitude information of the location in China.**
(CSV)

**S1 Table. Percent contribution of 19 bioclimatic variables.**
(DOCX)

**S2 Table. Pairwise Pearson' s correlation coefficients of environmental variables.**
(DOCX)

## Author Contributions

**Data curation:** Pan Jiang.

**Supervision:** Rulin Wang.

**Validation:** Yulin Yang.

**Visualization:** Yi Huang.

**Writing – original draft:** Jingtian Yang.

**Writing – review & editing:** Rulin Wang, Yuxia Yang.

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
