## [Decision Letter · Decision Letter 0]

23 Nov 2021

PONE-D-21-32985Prediction of Potential Geographic Distribution of Relict Plant Pteroceltis tatarinowii in China Based on MaxEntPLOS ONE

Dear Dr. Yuxia Yang,

Thank you for submitting your manuscript to PLOS ONE. After careful consideration, we feel that it has merit but does not fully meet PLOS ONE’s publication criteria as it currently stands. Therefore, we invite you to submit a revised version of the manuscript that addresses the points raised during the review process.

We look forward to receiving your revised manuscript.

Kind regards,

Randeep Singh

Academic Editor

PLOS ONE

Journal Requirements:

"JY and YH have been funded by the Scientific research initiation project of Mianyang Normal University (QD2019A13) and the Open Project from the Ecological Security and Protection Key Laboratory of Sichuan Province ( ESP1608 and ESP1801)"

5. Please amend your authorship list in your manuscript file to include author Rulin Wang.

6. We note that Figures 1, 6, 7 , 8 and 9 in your submission contain [map/satellite] images which may be copyrighted. All PLOS content is published under the Creative Commons Attribution License (CC BY 4.0), which means that the manuscript, images, and Supporting Information files will be freely available online, and any third party is permitted to access, download, copy, distribute, and use these materials in any way, even commercially, with proper attribution. For these reasons, we cannot publish previously copyrighted maps or satellite images created using proprietary data, such as Google software (Google Maps, Street View, and Earth). For more information, see our copyright guidelines: http://journals.plos.org/plosone/s/licenses-and-copyright.

a. You may seek permission from the original copyright holder of Figures 1, 6, 7 , 8 and 9 to publish the content specifically under the CC BY 4.0 license.  

Reviewers' comments:

Reviewer's Responses to Questions

**Comments to the Author**

1. Is the manuscript technically sound, and do the data support the conclusions?

Reviewer #1: Yes

Reviewer #2: Yes

2. Has the statistical analysis been performed appropriately and rigorously? 

Reviewer #1: Yes

Reviewer #2: Yes

3. Have the authors made all data underlying the findings in their manuscript fully available?

Reviewer #1: Yes

Reviewer #2: Yes

4. Is the manuscript presented in an intelligible fashion and written in standard English?

Reviewer #1: Yes

Reviewer #2: No

5. Review Comments to the Author

Reviewer #1: The MS is technically valid and author have used appropriate modeling techniques and validations criteria. Also the MS is written in proper English language. All the methods and analysis are described well.

Reviewer #2: Abstract seems to qualitatively stated. Pls speak numerically.

The authors predicted the future using the emission scenarios. How about the past? What happened during the calibration period?

The title says site suitability analysis for Pteroceltis tatarinowii. Should the paper only be concerned with this? The first sentence of the introduction gives me the impression that authors are evaluating the impact of climate change on the distribution of this plant. If so, the title and abstract need to change accordingly.

English usage needs serious improvement.

The authors need to provide a better literature review on Maxent application in china the results obtained. The advantages and disadvantages of this model should be provided in comparison with other similar models.

I believe the intro part for the Pteroceltis tatarinowii species could be briefed.

What is the innovation of this research?

The methods section could be improved. The readers are only concerned with the general flow of the work. Small details like saving a file in .CSV format is not relevant.

How did you identify the inaccurate sites in the distribution records of P. tatarinowii. What criteria did you use?

Figure 1 legend if not informative. What are those high and low values? If possible, use a graded symbology for the distribution points in terms of frequency. Larger circles indicating higher numbers of this plant. This way the reader can guess which part of China are more suitable for this plant. If the study area is the whole country of china, I believe the first part of the methods section must be concerned with describing the distribution of environmental factors across China.

“…scenarios could describe the prediction results of future climate change more scientifically.” Citation pls.

The terms used in table 1 should be clarified in the text.

It would be useful to have a flowchart for the analysis. The details can be provided in this chart instead of the text.

“… The closer the AUC ...simulation accuracy were divided into four grades: poor….” Citation pls.

Section 3.2 is the repetition of Fig 6. Pls provide only the results not noticeable on the map.

Based on your results in table 2, precipitation in China will increase significantly in the future under the climate change. The results in this table are only valid if there will be no more expansion of human activities in China which is highly unlikely.

A lot of the results are repeated in the discussion section. Pls only elaborate on your findings and avoid restating the results. This section has become long and tedious.

6. PLOS authors have the option to publish the peer review history of their article (what does this mean?). If published, this will include your full peer review and any attached files.

Reviewer #1: No

Reviewer #2: **Yes: **Masoud Jafari Shalamzari

---

## [Author Response · Author response to Decision Letter 0]

17 Dec 2021

Dear Editors and Reviewers: 

We would like to thank the editor for giving us a chance to resubmit our manuscript entitled “Prediction of Potential Geographic Distribution of Relict Plant Pteroceltis tatarinowii in China Based on MaxEnt” (ID: PONE-D-21-32985), and also thank the reviewers for giving us constructive suggestions which would help us both in English and in depth to improve the quality of the paper. Here we submit a new version of our manuscript with the title “Potential Geographic Distribution of Relict Plant Pteroceltis tatarinowii in China under Climate Change Scenarios”, which has been modified according to the reviewers’ suggestions. Efforts were also made to correct the mistakes and improve the English of the manuscript. We mark all the changes in red in the revised manuscript. The main corrections in the paper and the responds to the reviewer’s comments are as flowing: 

Responds to the comments of Editor：

1.Comment: We note that Figures 1, 6, 7 , 8, 9 and 10 in your submission contain [map/satellite] images which may be copyrighted. All PLOS content is published under the Creative Commons Attribution License (CC BY 4.0), which means that the manuscript, images, and Supporting Information files will be freely available online, and any third party is permitted to access, download, copy, distribute, and use these materials in any way, even commercially, with proper attribution. For these reasons, we cannot publish previously copyrighted maps or satellite images created using proprietary data, such as Google software (Google Maps, Street View, and Earth). For more information, see our copyright guidelines: http://journals.plos.org/plosone/s/licenses-and-copyright.

Response: The 1：16 million Chinese administrative division map was taken from the Ministry of Natural Resources of the People's Republic of China (http://bzdt.ch.mnr.gov.cn/index.html) (L146-L147). We confirm that there is no copyright issues, and they can be freely available online, and any third party is permitted to access, download, copy, distribute, and use these Figures in any way, even commercially, with proper attribution. 

Responds to the comments of Reviewer #2: 

1.Comment: Abstract seems to qualitatively stated. Pls speak numerically. 

Response: Thank you for your valuable advice. The abstract has been modified and stated numerically. (L35-L39). 

2.Comment: (The authors predicted the future using the emission scenarios. How about the past? What happened during the calibration period?) 

Response: Thank you for your valuable advice. In the discussion, we pointed out the significance of simulating past scenarios and the limitation of this study. (L405-L412).

.

3. Comment: The title says site suitability analysis for Pteroceltis tatarinowii. Should the paper only be concerned with this? The first sentence of the introduction gives me the impression that authors are evaluating the impact of climate change on the distribution of this plant. If so, the title and abstract need to change accordingly. 

Response: Thank you for your valuable advice. We revised the title to “Potential Geographic Distribution of Relict Plant Pteroceltis tatarinowii in China under Climate Change Scenarios” .

4. Comment: The authors need to provide a better literature review on Maxent application in china the results obtained. The advantages and disadvantages of this model should be provided in comparison with other similar models. 

Response: Thank you for careful work. We provide a better literature review in Introduction (L72-L81).

5.Comment: I believe the intro part for the Pteroceltis tatarinowii species could be briefed.. 

Response: Thank you for your careful work. As Reviewer suggested that the intro part for the Pteroceltis tatarinowii species was briefed (L82-L91).

6. Comment: What is the innovation of this research?. 

Response: As Reviewer suggested that the innovation of this research was supplemented in discussion part (L420-L442).

7. Comment: The methods section could be improved. The readers are only concerned with the general flow of the work. Small details like saving a file in .CSV format is not relevant.

Response: Thank you for your careful work. The methods section was improved according to reviewer’s comment (L127, L138-L139, L146, L155)

8.Comment: How did you identify the inaccurate sites in the distribution records of P. tatarinowii. What criteria did you use?

Response: The method to identify the inaccurate site was added in L120-L127

9.Comment: Figure 1 legend if not informative. What are those high and low values?

Response: Thank you for your careful work. The information in the legend of Figure 1 has been improved (Fig. 2)

10. If possible, use a graded symbology for the distribution points in terms of frequency. Larger circles indicating higher numbers of this plant. This way the reader can guess which part of China are more suitable for this plant. 

Response: As Reviewer suggested that a graded symbology for the distribution points in terms of frequency. (Fig. 2)

.

10. Comment: If the study area is the whole country of china, I believe the first part of the methods section must be concerned with describing the distribution of environmental factors across China. 

Response: Thank you for your valuable advice. As Reviewer suggested that the describing of climate factors across China was added in L105-L116.

11. Comment: “…scenarios could describe the prediction results of future climate change more scientifically.” Citation pls. 

Response: Thank you for your careful work. Citation was supplemented (L139).

12. Comment: The terms used in table 1 should be clarified in the text.

Response: Thank you for your careful work. We have clarified.the terms used in Table 1. 

13. Comment: It would be useful to have a flowchart for the analysis. The details can be provided in this chart instead of the text.. 

Response: .Thank you for your valuable advice. We have supplemented a flowchart. (Fig. 1)

14. Comment: “… The closer the AUC ...simulation accuracy were divided into four grades: poor….” Citation pls.

Response: Thank you for your careful work. Citation was added. (L168, L170)

15. Comment: Section 3.2 is the repetition of Fig 6. Pls provide only the results not noticeable on the map. 

Response: We have revised the expression of section 3.2 (L248-L249, L251-L253). 

16. Comment: Based on your results in table 2, precipitation in China will increase significantly in the future under the climate change. The results in this table are only valid if there will be no more expansion of human activities in China which is highly unlikely. 

Response: As Reviewer suggested that the influence of human activities was discussed in section 4.3 (L412-L419). 

17.Comment: A lot of the results are repeated in the discussion section. Pls only elaborate on your findings and avoid restating the results. This section has become long and tedious. 

Response: Thank you for your careful work. We revised the discussion part and deleted the repeated part. 

18.Comment: English usage needs serious improvement.

Response: Thank you for your careful work. We have made every effort to improve the use of English. 

We tried our best to improve the manuscript and made some changes in the manuscript. These changes will not influence the content and framework of the paper. And here we did not list the changes but marked in red in revised paper. 

We appreciate for Editors/Reviewers’ warm work earnestly, and hope that the correction will meet with approval. 

Once again, thank you very much for your comments and suggestions.

---

## [Decision Letter · Decision Letter 1]

17 Feb 2022

PONE-D-21-32985R1Potential Geographic Distribution of Relict Plant Pteroceltis tatarinowii in China under Climate Change ScenariosPLOS ONE

Dear Dr. Yang,

Thank you for submitting your manuscript to PLOS ONE. After careful consideration, we feel that it has merit but does not fully meet PLOS ONE’s publication criteria as it currently stands. Therefore, we invite you to submit a revised version of the manuscript that addresses the points raised during the review process.

Revise the manuscriptIndicate which changes you require for acceptance versus which changes you recommendAddress any conflicts between the reviews so that it's clear which advice the authors should followProvide specific feedback from your evaluation of the manuscriptPlease ensure that your decision is justified on PLOS ONE’s publication criteria and not, for example, on novelty or perceived impact.

We look forward to receiving your revised manuscript.

Kind regards,

Randeep Singh

Academic Editor

PLOS ONE

Reviewers' comments:

Reviewer's Responses to Questions

**Comments to the Author**

1. If the authors have adequately addressed your comments raised in a previous round of review and you feel that this manuscript is now acceptable for publication, you may indicate that here to bypass the “Comments to the Author” section, enter your conflict of interest statement in the “Confidential to Editor” section, and submit your "Accept" recommendation.

Reviewer #2: All comments have been addressed

Reviewer #3: All comments have been addressed

Reviewer #4: (No Response)

Reviewer #5: (No Response)

2. Is the manuscript technically sound, and do the data support the conclusions?

Reviewer #2: Yes

Reviewer #3: Yes

Reviewer #4: Yes

Reviewer #5: No

3. Has the statistical analysis been performed appropriately and rigorously? 

Reviewer #2: Yes

Reviewer #3: Yes

Reviewer #4: No

Reviewer #5: No

4. Have the authors made all data underlying the findings in their manuscript fully available?

Reviewer #2: Yes

Reviewer #3: Yes

Reviewer #4: No

Reviewer #5: Yes

5. Is the manuscript presented in an intelligible fashion and written in standard English?

Reviewer #2: Yes

Reviewer #3: Yes

Reviewer #4: No

Reviewer #5: No

6. Review Comments to the Author

Reviewer #2: I have reviewed the file and it seems that you have done a great effort in enhancing the quality of your paper. Thanks for considering my comments.

Wish you all the luck in your future endeavors.

Reviewer #3: As a unique single species genus in China, Pteroceltis tatarinowii is scattered or scattered in 19 provinces and regions in China. Due to the destruction of natural vegetation, it is often cut down in large quantities, resulting in the gradual reduction of the distribution area, the forest appearance is broken, and there are few residues in some areas. Therefore, the author uses MaxEnt, which is widely used at present, to simulate its distribution, which is of great significance for the resource protection of this tree species. The manuscript method is accurate, the result is reasonable, the discussion is sufficient, and has high scientific value. After review, I think the author has carefully revised the comments of the reviewers and reached the standard of publication. So my suggestion is to accept.

Reviewer #4: The manuscript title “Potential Geographic Distribution of Relict Plant Pteroceltis tatarinowii in China under Climate Change Scenarios” model the potential distribution of Pteroceltis tatarinowii using MaxEnt and 11 environmental variables. The manuscript is still some distance away from publication.

There are some errors need to address.

1. Page4, line 127: “second” this part might be needs a reference to confirm your method.

2. page4, line133：Please explain why choice environment variables from worldclim? Where you get the Al, aspect and slope?

3. page4, section 2.3: Considering the uncertainty of future climate scenarios, assessments of the impact should incorporate data from a range of climate models that represent different climate sensitivities to various possible future climate change projections. Thus, to reduce this impact on global circulation models (GCMs) on which the future climate datasets are based, an ensemble forecasting procedure was need to use in current study.

4. page5, line144：please list the result of Spearman correlation analysis in your manuscript. How to implement the Spearman correlation analysis? Which software? Please list in manuscript.

5. page5, section 2.4: please list the detail parameters of maxent were used in your study.

6. page5, section 2.4: which method were used to confirm the RM and feature? and how to implement? How to get the LQHPT?

7. page5, section 2.5: AUC have many disadvantages to assess the performance of model, especially for Maxent. Thus, the pROC, TSS or Kappa were used to assess the model.

8. page14, references: Please keep the reference format consistent. Thank you!

Reviewer #5: I read the MS with interest and I noticed that the authors tried to address all the points highlighted by the reviewer; however, I note again several critical points and even the English stile is critical and a thorough revision is necessary to make the text suitable for a scientific community.

Here are my main observations:

- I imagine that “absence points” (together with those of occurrence) were also used to build the model but nothing is said about this in the text. Please also clarify this aspect which is not mentioned in the text (how they were generated or directly detected).

- even the creation of the buffer does not convince me since it is a relict tertiary plant. why was it necessary to do this (and how does it relate to the altitude, slope and aspect reported in table 1?)

- all variables were imported into the MaxEnt and repeated 3 times (L141): pls justify this procedure.

- In the method section “223 effective distribution points of P. tatarinowii and 11 environmental variables” were considered, but in the results only “96 distribution points” were mentioned. What happened?

- L322: the authors refer of "the gravity center of P. tatarinowii's potential suitable habitat"; this treatment upsets me and I do not understand how it is possible to associate the potential habitat of a tertiary relict plant with a “center of gravity”. They seem to me to be two extremely different and unrelated concepts. My suspicion is that a good graphic picture of the model is being used to try to define something ecologically much more complex. So that, the authors refer to “potential geographical distribution (L388). The authors should clarify this fundamental aspect. Curiously, in fact, the authors declare (LL363-366) that “vegetation coverage, also have certain influence on the potential geographical distribution of P. tatarinowii. However, this variable is not included in the simulation because ... .. "but this is a primary trait that defines the habitat of the species. There is a lot of literature on this aspect which should be carefully considered.

- L418: one of the main results of this study is that “the future suitable area of P. tatarinowii simulated in this study is wider than the actual situation”. this seems counterintuitive to me: I expect a tertiary relict plant to be damaged by global change and not the other way around. Assuming that the model is accurate (as stated on the basis of its values) how can this ecological contradiction be explained? if it doesn't make sense I would think the model is beautiful but absolutely not predictive. In this section of “limitations” this point should be explored in detail because curiously other studies, in other contexts (see for example https://doi.org/10.1007/s10531-020-02029-y + https://doi.org/10.3390/d12040157), highlight that many models are considered without a critical evaluation while, verified in the field, they reveal only unrealistic simulations

- considering what has just been said, it does not seem to me that it can be stated that "Using SDMs to simulate the past, current and future changes of suitable habitat of relict plants is an effective method". Much precaution should be used on this issue.

I expect the authors to convince me of the soundness of their study.

Minor points:

- the authors speak of a rare species but I observe very large distribution areas and therefore we should speak only of threatened species (specifying what it is threatened by).

- LL54-57: this sentence is not clear and, indeed, in its present form it seems just wrong / contradictory.

- L58; pls add the year after IPCC

-LL72-73: …among others

- L82: add patronymic

-L86: I believe that the ecological interest prevails over the other two.

- L89: only one population?

- LL99-101: delete this sentence

-L118: the occurrences of

- Table 1: Altitude, aspect and slope mean values (on not)?

- L373: add year of publication

-L391: meta-analysis

-L405: not all tertiary relict plants are “living fossil plants”.

-LL407-413: add nothing to the MS and should be eliminated.

-L422: mini?

7. PLOS authors have the option to publish the peer review history of their article (what does this mean?). If published, this will include your full peer review and any attached files.

Reviewer #2: **Yes: **Masoud Jafari Shalamzari

Reviewer #3: No

Reviewer #4: No

Reviewer #5: No

---

## [Author Response · Author response to Decision Letter 1]

2 Mar 2022

Dear Editors and Reviewers: 

We would like to thank the editor for giving us a chance to resubmit our manuscript entitled “Prediction of Potential Geographic Distribution of Relict Plant Pteroceltis tatarinowii in China Based on MaxEnt” (ID: PONE-D-21-32985), and also thank the reviewers for giving us constructive suggestions which would help us both in English and in depth to improve the quality of the paper. Here we submit a new version of our manuscript with the title “Potential Geographic Distribution of Relict Plant Pteroceltis tatarinowii in China under Climate Change Scenarios”, which has been modified according to the reviewers’ suggestions. Efforts were also made to correct the mistakes and improve the English of the manuscript. We mark all the changes in red in the revised manuscript. The main corrections in the paper and the responds to the reviewer’s comments are as flowing: 

Responds to the comments of Reviewer #4: 

1.Comment: Page4, line 127: “second” this part might be needs a reference to confirm your method.

Response: Thank you for your valuable advice. A reference has been added to confirm my method (L127). 

2.Comment: Please explain why choice environment variables from worldclim? 

Response: Thank you for your valuable advice. We have explained the reasons for choosing to download variables from worldclim and supplemented the corresponding references. (L132-L134).

3.Comment: Where you get the Al, aspect and slope?

Response: The sources of altitude, slope and aspect data have been supplemented. (L135-L136).

.

4. Comment: page4, section 2.3: Considering the uncertainty of future climate scenarios, assessments of the impact should incorporate data from a range of climate models that represent different climate sensitivities to various possible future climate change projections. Thus, to reduce this impact on global circulation models (GCMs) on which the future climate datasets are based, an ensemble forecasting procedure was need to use in current study. 

Response: Thank you for your valuable advice. In L139-L140, the reasons for choosing BCC-CSM2-MR model have been supplemented..

5. Comment: page5, line144：please list your manuscript. How to implement the Spearman correlation analysis? Which software? Please list in manuscript. 

Response: We provide the result of Spearman correlation analysis in Table S1.

6.Comment: page5, section 2.4: please list the detail parameters of maxent were used in your study.

Response: Thank you for your careful work. As Reviewer suggested that the detail parameters of maxent was supplemented (L157-L162).

7. Comment: page5, section 2.5: AUC have many disadvantages to assess the performance of model, especially for Maxent. Thus, the pROC, TSS or Kappa were used to assess the model.

Response: As reviewer suggested that the we have explained this point in the discussion and added supporting literature. (L349-L357).

Responds to the comments of Reviewer #5: 

1. Comment: I imagine that “absence points” (together with those of occurrence) were also used to build the model but nothing is said about this in the text. Please also clarify this aspect which is not mentioned in the text (how they were generated or directly detected).

Response: An important feature of MaxEnt model is that only ‘no-absence points’ is required for modeling, which is supplemented in the introduction..(L75-L76)

2.Comment: even the creation of the buffer does not convince me since it is a relict tertiary plant. why was it necessary to do this (and how does it relate to the altitude, slope and aspect reported in table 1?

Response: Thank you for your valuable advice. The method for selecting distribution points was provided in L123-L127.

3.Comment: all variables were imported into the MaxEnt and repeated 3 times (L141): pls justify this procedure.

Response: Thank you for your careful work. We confirmed this procedure.

\\

8.4. Comment: In the method section “223 effective distribution points of P. tatarinowii and 11 environmental variables” were considered, but in the results only “96 distribution points” were mentioned. What happened?

Response: Thank you for your careful work. It should be 223 points (L127, L193)

.

5. Comment: L322: the authors refer of "the gravity center of P. tatarinowii's potential suitable habitat"; this treatment upsets me and I do not understand how it is possible to associate the potential habitat of a tertiary relict plant with a “center of gravity”. They seem to me to be two extremely different and unrelated concepts. My suspicion is that a good graphic picture of the model is being used to try to define something ecologically much more complex. So that, the authors refer to “potential geographical distribution (L388). The authors should clarify this fundamental aspect. Curiously, in fact, the authors declare (LL363-366) that “vegetation coverage, also have certain influence on the potential geographical distribution of P. tatarinowii. However, this variable is not included in the simulation because ... .. "but this is a primary trait that defines the habitat of the species. There is a lot of literature on this aspect which should be carefully considered.

Response: Thank you for your valuable advice. As Reviewer suggested that we have supplemented the research methods and references of gravity center in the method part.(L187-L189).

6. Comment: the authors speak of a rare species but I observe very large distribution areas and therefore we should speak only of threatened species (specifying what it is threatened by).

. 

Response: Thank you for your careful work. After our verification, P. tatarinowii is a unique relic plant in China.

7. Comment: L54-57: this sentence is not clear and, indeed, in its present form it seems just wrong / contradictory.

Response: We have rewritten this sentence (L54-L55) 

8. Comment: L58; pls add the year after IPCC

Response: We have added the year after IPCC(L57)

9. Comment: L72-73: …among others

Response: As reviewer suggested that we have corrected the sentence (L72)

10. Comment: - L82: add patronymic 

Response: We have add patronymic (L81). 

11. Comment: -L86: I believe that the ecological interest prevails over the other two. 

Response: As Reviewer suggested that we have rewritten this sentence (L85). 

12.Comment: L89: only one population? 

Response: Thank you for your careful work. We have corrected it (L88). 

13.Comment: L99-101: delete this sentence

Response: Thank you for your valuable advice. We have deleted this part. 

14.Comment: L118: the occurrences of

Response: We have corrected it according to reviewer’s advice (L116) 

15.Comment:L373: add year of publication

Response: We have added year of publication in L393.

16.Comment: L391: meta-analysis

Response: We have corrected in L410.

17.Comment: L405: not all tertiary relict plants are “living fossil plants”.

Response: Thank you for your careful work. We have corrected in L424.

18.Comment: L422: mini?.

Response: Thank you for your careful work. We have corrected it in L440.

19.Comment: L418: one of the main results of this study is that “the future suitable area of P. tatarinowii simulated in this study is wider than the actual situation”. this seems counterintuitive to me

Response: We explained this conclusion in the discussion (L435-L437).

We tried our best to improve the manuscript and made some changes in the manuscript. These changes will not influence the content and framework of the paper. And here we did not list the changes but marked in red in revised paper. 

We appreciate for Editors/Reviewers’ warm work earnestly, and hope that the correction will meet with approval. 

Once again, thank you very much for your comments and suggestions.

---

## [Editor Report · Decision Letter 2]

15 Mar 2022

Potential Geographic Distribution of Relict Plant Pteroceltis tatarinowii in China under Climate Change Scenarios

PONE-D-21-32985R2

Dear Dr. Yang,

We’re pleased to inform you that your manuscript has been judged scientifically suitable for publication and will be formally accepted for publication once it meets all outstanding technical requirements.

Kind regards,

Randeep Singh

Academic Editor

PLOS ONE
---

## [Editor Report · Acceptance letter]

30 Mar 2022

PONE-D-21-32985R2 

Potential Geographic Distribution of Relict Plant *Pteroceltis tatarinowii* in China under Climate Change Scenarios 

Dear Dr. Yang:

I'm pleased to inform you that your manuscript has been deemed suitable for publication in PLOS ONE. Congratulations! Your manuscript is now with our production department. 

Kind regards, 

on behalf of

Dr. Randeep Singh 

Academic Editor

PLOS ONE